# Wearable Device for Continuous and Real-Time Monitoring of Human Sweat Sodium

**DOI:** 10.3390/s25113467

**Published:** 2025-05-30

**Authors:** Anas Mohd Noor, Muhammad Salman Al Farisi, Mazlee Mazalan, Nur Fatin Adini Ibrahim, Asnida Abdul Wahab, Zulkarnay Zakaria, Nurul Izni Rusli, Norhayati Sabani, Asrulnizam Abd Manaf

**Affiliations:** 1Faculty of Electronic Engineering & Technology, Universiti Malaysia Perlis, Arau 02600, Malaysia; mazleemazalan@unimap.edu.my (M.M.); fatinadini@studentmail.unimap.edu.my (N.F.A.I.); zulkarnay@unimap.edu.my (Z.Z.); nurulizni@unimap.edu.my (N.I.R.); hayatisabani@unimap.edu.my (N.S.); 2Centre of Excellence for Sports Engineering Research Center, Universiti Malaysia Perlis, Arau 02600, Malaysia; 3Department of Biomedical Information Sciences, Hiroshima City University, Hiroshima 731-3194, Japan; 4Centre of Excellence for Micro System Technology, Universiti Malaysia Perlis, Arau 02600, Malaysia; 5Department of Biomedical Engineering and Health Sciences, Faculty of Electrical Engineering, Universiti Teknologi Malaysia, Johor Bahru 81310, Malaysia; asnida.aw@utm.my; 6Collaborative Microelectronic Design Excellence Center, Universiti Sains Malaysia, Bayan Lepas 11900, Malaysia; eeasrulnizam@usm.my

**Keywords:** microfluidic chip, sweat sodium measurement, wearable device

## Abstract

Wearable sweat-sensing devices hold significant potential for non-invasive, continuous health monitoring. However, challenges such as ensuring data accuracy, sensor reliability, and measurement stability persist. This study presents the development of a wearable system for the real-time monitoring of human sweat sodium levels, addressing these challenges through the integration of a novel microfluidic chip and a compact potentiostat. The microfluidic chip, fabricated using hydrophilic materials and designed with vertical channels, optimizes sweat flow, prevents backflow, and minimizes sample contamination. The developed wearable potentiostat, as a measurement device, precisely measures electrical currents across a wide dynamic range, from nanoamperes to milliamperes. Validation results demonstrated accurate sodium concentration measurements ranging from 10 mM to 200 mM, with a coefficient of variation below 4% and excellent agreement with laboratory instruments (intraclass correlation = 0.998). During physical exercise, the device measured a decrease in sweat sodium levels, from 101 mM to 67 mM over 30 min, reflecting typical physiological responses to sweating. These findings confirm the system’s reliability in providing continuous, real-time sweat sodium monitoring. This work advances wearable health-monitoring technologies and lays the groundwork for applications in fitness optimization and personalized hydration strategies. Future work will explore multi-biomarker integration and broader clinical trials to further validate the system’s potential.

## 1. Introduction

Wearable sweat-sensing devices are transforming health monitoring by offering non-invasive, real-time analysis of physiological and biochemical parameters. Sweat, as a biofluid, contains various biomarkers, including electrolytes, metabolites, and hormones, which provide critical insights into hydration status, electrolyte balance, and overall health [1,2]. Unlike traditional blood tests, which are invasive and pose risks such as skin inflammation, wearable sweat sensors enable continuous monitoring, making them more convenient and user-friendly [3]. These systems have broad applications, from helping athletes optimize performance and prevent dehydration by tracking electrolyte loss during exercise to assisting individuals with chronic medical conditions, such as diabetes, in monitoring biomarkers such as sweat glucose and reducing the need for frequent blood draws. Physicians also benefit from these devices, as the continuous data stream supports more informed diagnoses and treatment plans [4].

Wearable sweat-sensing systems typically consist of three main components: (1) a sweat collection module, (2) sensors for biomarker detection, and (3) an electrochemical interface, such as a potentiostat, which converts chemical signals into measurable electrical outputs for real-time monitoring [5,6]. However, effective sweat collection remains a challenge, as sweat secretion is limited to microliter volumes. Identifying optimal body locations for efficient sweat collection is critical to device performance. Sweat-mapping studies have revealed that the forehead produces the highest sweat rates, followed by the anterior and posterior torso, hands, and legs [7,8]. These findings guide the design of sweat collection devices capable of capturing limited sweat volumes effectively [9]. However, challenges such as power efficiency, data security, material stability, and mass production hinder widespread adoption. Future advancements aim to integrate artificial intelligence, enhance multifunctionality, and improve interoperability, paving the way for personalized, remote healthcare solutions [10,11].

Microfluidic chips or devices have emerged as promising solutions for sweat collection and analysis. These chips offer significant advantages, such as reducing evaporation, minimizing interference from skin debris, and enabling real-time biomarker analysis through integrated channels [12,13,14,15]. Despite these benefits, existing designs face limitations in controlling sweat flow rates. High flow rates can lead to rapid discharge of sweat before sensors can capture the data, while low flow rates may cause the mixing of old and new samples, compromising data accuracy [16,17,18]. Material properties also pose challenges. Many microfluidic chips are made from hydrophobic materials, which hinder efficient fluid flow. Although surface treatments can improve hydrophilicity, these modifications often degrade over time, reverting to hydrophobicity due to environmental exposure [19,20]. Consequently, robust hydrophilic materials that maintain consistent fluid flow over extended periods are needed to enhance device performance [21].

Microfluidic methods utilizing passive pumps, such as gravity and capillary forces, offer cost-effective and energy-efficient fluid manipulation. These methods rely on natural forces to move fluids through microchannels without requiring external power sources. Gravity-driven flow uses the force of gravity to drive fluid, while capillary forces exploit the fluid’s tendency to move through narrow channels. These passive approaches are ideal for wearable devices, enabling continuous and real-time monitoring, such as sweat collection [22]. In contrast, active pumping methods use external pumps to control fluid flow, offering more precise flow regulation. While these systems are versatile and allow for accurate flow control, they require additional components, increasing system complexity and power consumption. For sweat collection, passive methods are advantageous because they simplify design and reduce energy requirements [23,24]. However, both passive and active methods face challenges. Variations in sweat rate, skin conditions, and environmental factors can impact accuracy. Active systems offer better flow control but are more complex and power-intensive, while passive systems may face limitations in sensitivity and consistent flow. The choice depends on balancing efficiency, complexity, and specific application needs.

Beyond fluid handling, wearable sweat-sensing devices must achieve high accuracy and selectivity. High accuracy is essential for detecting specific analytes, such as sodium, particularly at low concentrations [25,26,27]. Selectivity is equally important to differentiate target ions such as sodium from interfering ions such as potassium and calcium, ensuring reliable data in the complex composition of sweat [23,28]. Additionally, maintaining stability and consistency in electrical signal measurements over prolonged use remains a significant challenge, as external factors such as temperature changes and user movement can introduce variability [29,30].

This study addresses current limitations in wearable sweat sensing by developing a novel platform for real-time sodium monitoring with three key advancements: (1) a 3D-printed hydrophilic microfluidic chip featuring vertical inlets to ensure continuous sweat flow, prevent backflow, and minimize contamination—overcoming the challenges of flow control and material hydrophobicity in existing designs; (2) integration with our previously developed We-VoltamoStat [31], a compact, Bluetooth-enabled potentiostat that achieves lab-grade accuracy in a wearable form factor; and (3) a complete system enabling real-time wireless monitoring during physical exercise, demonstrated by tracking physiological sodium dynamics. By combining these innovations in fluid handling, sensor accuracy, and practical usability, the platform significantly advances wearable health-monitoring technologies for both fitness and clinical applications.

## 2. Materials and Methods

### 2.1. Fabrication of Microfluidic Chip

The design was chosen to leverage gravity-driven fluid dynamics, ensuring uninterrupted sweat flow while preventing backflow. This approach minimizes the mixing of old and new sweat samples, thereby enhancing the accuracy of real-time measurements. Additionally, the vertical channel design allows for efficient use of space within the wearable device, making it compact and user-friendly. Water-washable resin is chosen to enhance the wettability properties of microfluidic chips, which are crucial for fluid flow applications, especially in microfluidic chips. In contrast to standard non-washable resin, which typically exhibits low wettability and high contact angles (hydrophobic), water-washable resin offers improved hydrophilicity.

The microfluidic chip was fabricated using a Creality LD-002H SLA 3D printer (Micro Center, Hilliard, OH, USA). The device’s geometry was designed in SolidWorks 2020, sliced with Chitubox software, and printed using a clear, UV-curable, water-washable resin (E-Sun, W100). Post-processing included cleaning, curing, and assembly. The microfluidic chip, shown in Figure 1, displays the schematic drawing of the chip and an example of the printed chip. Magnets were incorporated into the design to enable easy and secure attachment or removal of the sensor. The Horiba ISE sodium sensor (B722, Laqua Twin Sensor, Horiba, Kyoto, Japan) was selected for its compact size and reliable performance, as shown in Figure 1. The sensor casing was dismantled, leaving only the sensor part, where the electrode connection is used to read the current. The microfluidic chip collects sweat droplets from the inlet and directs them to the sensing area. The sensor holder part includes a sensor holder secured by magnets, a lid hinge for the sensor, holes for wires, and a band for holding the wearable microfluidic chip.

Eight inlets were incorporated into the microfluidic chip to maximize sweat collection within a short time. The inlets were designed with a channel size width of 1.8 mm to enhance capillary force and maintain a good flow while reducing contamination. The microchannel was designed for vertical fluid flow driven by gravity. To control the flow rate of sweat reaching the outlet and ensure the adequate filling of the sensing area, an outlet with a width and height of less than 1 mm was designed. This small outlet channel increases the time required for sweat to flow out due to the high-pressure difference between the large sensing area and the small outlet channel [32]. Additionally, the design is based on hydrostatic principles, requiring sweat to flow from a lower position to a higher position. This creates increased pressure at the outlet, which generates a back pressure that opposes the fluid outflow, reducing and controlling the sweat flow rate.

To optimize the printing process and ensure high-quality fabrication, we carefully adjusted the printing recipe by conducting multiple trials and testing different parameter values, focusing on slicing settings. The bottom layer requires a time exposure of 12 s, while the normal layer needs only 2.5 s, optimizing the curing process for different layers. The lift distance for both bottom and normal layers is set to 5 mm, maintaining uniform detachment from the resin vat. The speed during operation is also consistent, with both the bottom and normal layers having a lift speed of 50 mm/min. The retraction speed is adjusted for efficiency, with the bottom layer retracting at 150 mm/min and the normal layer at 190 mm/min. Optimizing these parameters was crucial for achieving precise dimensions, smooth surfaces, and consistent device performance. By fine-tuning the slicing settings, we were able to minimize defects and improve the overall quality of the fabricated microfluidic chip. Table 1 shows the 3D printer settings used to achieve the chip dimensions.

### 2.2. Microfluidic Chip Flow Test

The performance of the printed microfluidic chip was assessed through two primary tests: a hydrophilicity test and an efficiency test for fluid delivery to the sensor channel. The hydrophilicity test focused on evaluating the wettability of the microfluidic chip, fabricated with a water-washable resin, by measuring the contact angle of its surface. A contact angle of less than 90°, ideally below 60°, is considered optimal for promoting efficient fluid flow [30]. Meanwhile, the efficiency test examined the device’s ability to deliver sweat to the sensor channel. Several parameters were evaluated, including the volume of the channel, the time taken for sweat to flow out of the outlet, and the time required to completely fill the sensing area. The channel volumes were defined using SolidWorks and remained constant for the inlets and the sensing area, while the outlet channel sizes varied. The results indicated that the volume of the outlet channel significantly affected the time required for sweat to exit. To ensure the complete filling of the sensing area, the time for the fluid to exit the outlet was measured while monitoring the fluid level within the sensing area, as shown in Figure 2.

### 2.3. Performance Comparison of Developed We-Voltamostat Device with Standard Instrument

The current measurement capabilities of the We-VoltamoStat were evaluated through a calibration test using a photodiode and a current source (Keithley 6221, Keithley Instruments, Inc., Cleveland, OH, USA). The photodiode was illuminated by a light source, and the current source generated a range of currents, as shown in Figure 3. Both the We-VoltamoStat and the Agilent multimeter (Agilent 34401) measured the generated currents. By comparing the readings from both instruments across the range of currents, the accuracy and precision of the We-VoltamoStat were assessed. The photodiode provided an independent and reliable reference, ensuring the accuracy of the calibration process.

### 2.4. Dummy Cell Test

A dummy cell was used to test the electronic system of the We-VoltamoStat and verify its ability to accurately detect and resolve potentiostat issues. The dummy cell creates a stable electrochemical environment, essential for evaluating the device’s performance [33]. Based on the approach outlined by Caux et al., a simple series circuit was used, consisting of a 1 kΩ resistor and a 1000 µF capacitor [34]. However, the specific configuration of the dummy cell can vary depending on the potentiostat setup. On the other hand, in a commercial potentiostat, such as the Metrohm, the dummy cell is connected to the working electrode (WE) through a resistor, while the counter electrode (CE) and reference electrode (RE) are connected to a capacitor, as shown in Figure 4. In contrast, the We-VoltamoStat uses a simplified two-electrode configuration, where the WE is connected to a resistor and the CE to a capacitor. While this configuration is adequate for current measurement, it is important to note that a three-electrode configuration can offer more precise results in certain applications.

### 2.5. Accuracy Test

Several tests were conducted to evaluate the accuracy, selectivity, and stability of the We-VoltamoStat and sensor in detecting sodium ions. These tests involved analyzing samples with known sodium ion concentrations as artificial sweat, ranging from 10 mM to 200 mM [35]. The artificial sweat is prepared using a standard process involving sodium chloride (NaCI), potassium chloride (KCI), and calcium chloride (CaCI_2_), as these are the major components of sweat ions. Combinations of NaCl solution, KCl solution, and CaCl_2_ solution were also mixed in a volume ratio of 2:1:1, respectively, aligning with the typical higher volume and dominance of NaCl compared to other interfering ions commonly found in sweat. To assess the accuracy of the We-VoltamoStat in measuring sodium ions, we conducted a series of amperometry tests using a sodium sensor. The tests involved comparing the electrical currents measured by the We-VoltamoStat to those measured by a commercial potentiostat, the Metrohm instrument (µStat-i 400s) (Metrohm, Herisau, Switzerland). For each sodium concentration in artificial sweat, ranging from 10 mM to 200 mM, we performed amperometry measurements for 300 s. The resulting current data were analyzed to evaluate the graphical similarities and the percentage error of average electrical currents between the two devices. To assess the statistical significance of the differences between the two devices, we conducted a Bland–Altman plot analysis, a paired t-test, and calculated the intraclass correlation coefficient (ICC) and coefficient of variation (CV). The Bland–Altman plot illustrates the agreement between the devices, identifying any systematic bias or outliers. The paired t-test determined if there are statistically significant differences in the mean currents measured by the two devices. The ICC and CV were used to evaluate the reliability and consistency of the measurements.

### 2.6. Selectivity and Stability Test

The selectivity for sodium ions over potassium and calcium was ensured using ion-selective electrodes (ISEs) optimized for sodium. These electrodes utilize a membrane composition specifically designed to preferentially interact with sodium ions, thereby minimizing interference from other ions commonly found in sweat. In the selectivity tests, solutions containing varying concentrations of sodium, potassium, and calcium were prepared to simulate real sweat samples. The electrode’s response to each ion was measured and compared against a reference curve generated for sodium. The statistical analyses, including graphical representation and two-way ANOVA with Tukey’s HSD test, demonstrated significant differences between the sodium responses and those of interference ions. The results indicated a mean difference exceeding 100 nA for potassium and calcium, which confirms the high specificity of the sensor for sodium ions. This differentiation ensures accurate sodium measurement even in the presence of interfering ions. The tests were repeated with a mixture of all three ions to evaluate real-world scenarios. The consistent performance of the sensor under these conditions validates its robustness and applicability for the continuous monitoring of sweat sodium levels in practical applications.

To assess the stability and reproducibility of the amperometry measurements, we conducted multiple repetitions for each sweat sample concentration and each 10 min exercise session. Five measurements were performed for each condition, allowing us to calculate the standard deviation of the measured currents. A standard deviation (SD) value less than 10 was considered indicative of high stability in the measurements [36]. By repeating the experiments, we were able to evaluate the consistency of the device’s performance over time and under varying conditions.

### 2.7. Sweat Sodium Measurement During Physical Exercise

Human volunteer testing was conducted to evaluate the wearable device’s performance in real-world conditions, focusing on continuous flow and real-time measurement of sweat sodium levels. This study, which involved a single volunteer, was approved by the institutional ethics procedure. The microfluidic chip was placed on the subject’s forehead, a region known for its high sweat production, as shown in the experimental setup in Figure 5. This setup allowed for the wireless monitoring of sweat sodium levels, with real-time data transmitted to a smartphone app for convenient analysis. The volunteer followed a standardized exercise protocol, running on a treadmill at a speed of 15 km/h for 10 min, followed by a 3 min rest period. Sweat was collected at 10 min intervals using the microfluidic chip over a total duration of 30 min. Amperometry measurements were conducted at 0 V on the collected sweat samples. The testing protocol was repeated ten times over 10 consecutive days. Sweat sodium concentrations were estimated using the determined reference curve, providing reliable and repeatable measurements. The conversion from measured electrical current (I) to sodium ion concentration in an amperometric sensor is performed using the general empirical calibration formula, as shown in Equation (1). This curve represents a mathematical relationship derived from experimental measurements of current at known sodium concentrations.(1)$$\left[{Na}^{+}\right]=f\left(I\right)=aI^{n}+bI^{n-1}+\ldots +c$$
where

[Na^+^] is the sodium ion concentration (typically expressed in mmol/L);I is the measured current (in nA or μA);a, b, c are empirically determined calibration coefficients;n is the degree of the polynomial fit (commonly 1 for a linear model, or 2–3 for nonlinear models).

**Figure 5 sensors-25-03467-f005:**
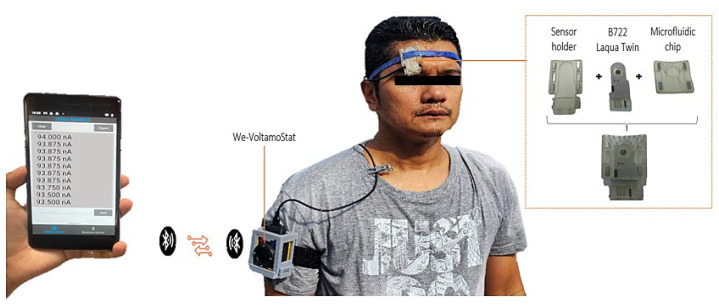
Wearable microfluidic sweat-sensing system during real-time monitoring. The microfluidic chip is worn on the forehead, with data transmitted wirelessly to a smartphone app for continuous sodium analysis.

## 3. Results and Discussion

### 3.1. Flow Test

Figure 6 shows the surface of the printed microfluidic chip, which was fabricated using a commercial water-washable resin (eSUN, Shenzhen, China). To evaluate the hydrophilicity of the surface, a water droplet was placed on the chip, and its contact angle was measured. The chip demonstrated excellent hydrophilicity with a contact angle of less than 45°. In comparison, a chip made with commercial conventional photopolymer resin (eSUN, W100, China) exhibited a contact angle greater than 90°, indicating a hydrophobic surface. The superior hydrophilic properties of the water-washable resin promote better wetting and smoother fluid flow within the microfluidic channels, significantly enhancing the device’s performance. Figure 7 shows the microfluidic chip fabricated in early 2023. A hydrophilicity test conducted more than two years later, using the same fabrication materials, shows that the contact angle remains below 45°. The exact angle measured is approximately 25°, determined using ImageJ 1.54f. This confirms that the device maintains its hydrophilic properties over time, demonstrating stable and long-lasting wettability.

### 3.2. Calibration Test

The We-VoltamoStat exhibited strong comparability to the Metrohm instrument in measuring electrical currents across varying voltage ranges. Both devices produced nearly identical current-voltage curves in cyclic voltammetry mode at a scan rate of 25 mV/s, as shown in Figure 8. This close alignment underscores the high accuracy and reliability of the We-VoltamoStat for electrochemical analysis. The resulting current-potential curve displayed a characteristic decay loop, which aligns with patterns reported in earlier studies [37,38]. This loop shape arises from the exponential decay of current spikes within the dummy cell, a well-documented phenomenon of resistor–capacitor (RC) circuits. A detailed comparative analysis of this wearable device against the bench-top instrument will be presented in the next sections.

### 3.3. Validation of the We-Voltamostat, Including Accuracy, Selectivity, and Stability

#### 3.3.1. Accuracy

The We-VoltamoStat consistently measured slightly lower current readings compared to the Metrohm instrument, as illustrated in Figure 9. This discrepancy is primarily attributed to the high current offset setting in the We-VoltamoStat, which can introduce noise and obscure certain current data points, resulting in a slight underestimation of the measured values. Despite this limitation, the We-VoltamoStat demonstrated remarkable accuracy in amperometry measurements, producing current-voltage graphs that closely aligned with those of the Metrohm instrument, as previously shown in Figure 9. The calculated percentage errors for average currents across various sodium chloride concentrations were minimal: 13.7% for 10 mM, 9.9% for 50 mM, 4.2% for 100 mM, 7.8% for 150 mM, and 8.8% for 200 mM. These findings highlight the We-VoltamoStat’s exceptional precision and reliability in current measurements, even when accounting for the influence of the current offset.

Table 2 presents the results of the sodium analysis tests, which evaluated the accuracy of the We-VoltamoStat in measuring sodium ion concentrations ranging from 1 mM to 250 mM. The calculated percentage errors for average currents were generally low, with values ranging from 4.2% to 13.7%. While lower concentrations (10 mM and below) exhibited slightly higher percentage errors, the overall performance of the We-VoltamoStat was satisfactory. The Bland–Altman plot in Figure 9 reveals an acceptable level of agreement between the We-VoltamoStat and the reference device for all concentrations.

The slight measurement discrepancies observed between the We-VoltamoStat and the Metrohm instruments are attributed to the high current offset setting in the We-VoltamoStat. This offset can mask subtle current data points, particularly at low concentrations, leading to an underestimation of the measured currents. While these differences were minor (mean difference of ~11.44 nA), they are within an acceptable range for wearable sensors and do not compromise practical applications. The We-VoltamoStat’s strong agreement with the Metrohm instrument, as confirmed by a high intraclass correlation coefficient (ICC) value of 0.998, ensures its reliability for real-world use. For practical applications, this level of precision is sufficient for monitoring sweat sodium in dynamic conditions such as exercise, where relative changes are more critical than absolute values.

Although there are higher percentage errors in current measurements at 1 mM and 10 mM, the Bland–Altman plot (Figure 9) shows that the We-VoltamoStat agrees well with the reference device across all concentrations, from 1 mM to 250 mM. The We-VoltamoStat consistently shows a small bias, approximately 11.44 nA lower than the reference device. This bias is more noticeable at lower concentrations, where the percentage errors are larger. However, the mean difference between the two devices for low concentrations (1 mM and 10 mM) is consistent with this expected bias. The higher errors at low concentrations are due to challenges in measuring very small sensor currents, where background noise can also contribute to the accuracy. Overall, the Bland–Altman plot confirms that the We-VoltamoStat delivers reliable measurements of sodium ions over a wide concentration range, even at low current values.

The statistical analyses comparing the We-VoltamoStat with the Metrohm instrument are shown in Table 3. The small standard error of the mean (SEM) of 1.469 nA reflects minimal measurement uncertainty, underscoring the device’s precision [39]. An ICC value of 0.998 indicates a high level of agreement between the two devices. The low coefficient of variation (CV) of 3.5% suggests consistent results across specific concentrations. Although the *p*-value of 0.0014 slightly exceeds the 0.001 significance threshold, it still points to no significant differences between the instruments, confirming the We-VoltamoStat’s comparable performance.

Figure 10 shows the relationship between the sensor current and sodium concentration during continuous sweat sample flow conditions. As the NaCl concentration decreased from 200 mM to 10 mM, the sensor current also decreased, reflecting the reduction in sweat sodium concentration. The average current readings from the We-VoltamoStat closely followed a polynomial curve, which can be used to estimate unknown sodium concentrations. The sensor’s current ranged from 37 to 135 nA for sodium concentrations between 10 mM and 200 mM. The average currents recorded by the We-VoltamoStat and the Metrohm instruments over five repetitions of amperometry are shown. Both instruments showed a strong correlation between current readings and sodium concentration, with the curves closely aligning. The results demonstrated that the sensor can be effectively modeled with a polynomial curve for sodium concentrations ranging from 1 mM to 250 mM. This highlights the sensor’s versatility and accuracy in measuring the sodium-related current across a broad range. Overall, these results demonstrate that the sensor is reliable, sensitive, and capable of accurately measuring sodium concentrations in sweat across a wide concentration range.

#### 3.3.2. Selectivity and Stability

Figure 11 shows the sensor’s ability to distinguish sodium ions from interfering ions such as potassium chloride (KCl) and calcium chloride (CaCl_2_) across a wide concentration range. This demonstrates the sensor’s strong selectivity for sodium ions and its resistance to interference from other common ions in sweat. The close match between the measured currents and the reference curve indicates minimal interference from KCl and CaCl_2_, even at higher concentrations. These findings confirm the sensor’s reliability in measuring sodium ions in real sweat samples.

The sodium-selective sensor evaluated in this study is based on a cation-selective membrane that operates according to the Nernstian principle. This membrane is engineered to respond specifically to monovalent and divalent cations (e.g., Na⁺, K⁺, Ca^2+^) through the incorporation of selective ionophores, enabling it to differentiate sodium ions from other ionic species. Importantly, due to its anionic nature, Cl^−^ does not participate in the ion-exchange process across the membrane, thereby reducing the likelihood of interference in sodium ion detection [40].

Table 4 presents the mean differences between sodium ions and mixed ions (Na⁺, K⁺, Ca^2+^) for concentrations ranging from 10 mM to 200 mM. The results further highlight the sensor’s high selectivity for sodium ions, as the mean differences for K⁺ and Ca^2+^ are significantly higher (over 100 nA) than those observed for sodium ions. While K⁺ ions typically exhibit negative deviations from the reference curve, Ca^2+^ ions contribute to positive current values, especially at concentrations exceeding 100 mM. However, considering the typical calcium concentration in sweat (0.07–12 mM), the degree of interference in sodium ion measurement remains minimal [41]. The amperometric measurements, repeated five times, exhibited consistently low standard deviations (SD) of less than 7 nA, indicating stable sensor performance. Moreover, the small standard error of the mean (SEM), typically less than 5 nA, further underscores the accuracy and consistency of the measurements.

### 3.4. Real-Time Sweat Sodium Monitoring

In this study, the developed wearable device was used to stream real-time current data from the Na⁺-selective ISE sensor during human testing. While the conversion of current values to Na⁺ concentration (in mmol/L) is performed offline using a predefined calibration curve, the measured current is directly correlated with sodium concentration and provides real-time insight into relative changes in Na⁺ levels. Two related studies demonstrated the feasibility of wearable electrochemical sensors for real-time sweat monitoring, with sodium concentrations determined offline. Schazmann et al. developed a Sodium Sensor Belt (SSB) using a sodium-selective electrode and fabric-based sweat collection to stream real-time signals, later converted to concentration via calibration [42]. Similarly, Pirovano et al. introduced the SwEatch platform, combining solid-contact ISEs with a 3D-printed microfluidic system and Bluetooth transmission, with offline conversion of signals to sodium levels [43].

Figure 12 illustrates the changes in sweat current (measured in nA) over a 30 min exercise session, with data collected at three distinct 10 min intervals: 0–600 s, 1200–1800 s, and 2400–3000 s. During these exercise periods, the sweat current showed a gradual decline, indicating a reduction in sweat sodium concentration over time. Physiologically, increased sweat rates are often linked to reduced sodium reabsorption efficiency in the sweat ducts, which can lead to higher sodium concentrations in sweat [44]. However, our study observed a progressive decline in sodium levels during the exercise phases, as reflected by the decreasing current. This trend is consistent with previous findings by Xu et al., who reported that subject 1 exhibited a reduction in sweat sodium concentration during low-intensity exercise. In contrast, subject 2 in the same study demonstrated an opposite trend, underscoring the high degree of inter-individual variability in sweat composition [45]. Furthermore, Pirovano et al. observed that the sodium concentration in sweat rose to 2.97 mM at approximately 34 min, before decreasing to 2.21 mM at around 58 min and further to 0.61 mM by 78 min [43]. Such discrepancies in sweat sodium profiles can be attributed to multiple physiological and environmental factors, including individual fitness level, heat acclimatization status, hydration practices, dietary sodium intake, genetics, and the functional responsiveness of eccrine sweat glands. These findings reinforce the notion that sweat composition is highly individualized, and that the accurate interpretation of sweat biomarkers necessitates careful consideration of both intrinsic and extrinsic variables [46].

### 3.5. Discussion

This study presents a novel wearable sweat-sensing platform that addresses key limitations in existing sodium monitoring systems. The integrated platform combines a sodium-selective ion-selective electrode (ISE), a Bluetooth-enabled miniaturized potentiostat (We-VoltamoStat), and a microfluidic chip with vertically oriented, gravity-assisted channels. Together, these components enable the real-time streaming of current signals during exercise, with sodium concentrations estimated offline via a predefined calibration curve. In comparison to previous wearable systems such as those developed by Schazmann et al. [42] and Pirovano et al. [43], which employed similar offline data processing approaches, the current system advances the field by integrating a more compact, low-power, and accurate electrochemical acquisition module.

The microfluidic chip further distinguishes this work through its vertical fluid transport design, which enhances flow stability and minimizes sample mixing. Fabricated using a water-washable photopolymer resin, the chip maintains long-term hydrophilicity, as demonstrated by sustained low contact angles (<45°) over a year after fabrication. This eliminates the need for post-fabrication surface treatments commonly required for PDMS-based systems, which often suffer from hydrophobic recovery within a day [46]. The use of a passive, gravity-driven system with a controlled outlet geometry also prevents backflow and ensures that fresh sweat is consistently delivered to the sensing area, addressing key challenges noted in earlier microfluidic designs [22].

An important physiological observation from this study is the progressive decline in sweat sodium concentration during the 30 min exercise session, as measured by decreasing current values. This phenomenon is consistent with established physiological mechanisms, in which sweat glands become more efficient at sodium reabsorption over time under the influence of aldosterone, particularly during prolonged or repeated sweating episodes [44,45]. As the body attempts to conserve electrolytes, sodium is increasingly reabsorbed from the primary sweat fluid in the duct before reaching the skin surface. This leads to lower sodium levels in the collected sweat over time, a trend also reported by Xu et al. [45] in human sweat analysis studies during exercise. Additionally, inter-individual variability in sweat electrolyte profiles—driven by factors such as fitness level, acclimatization, and glandular response—may further explain observed differences in sodium dynamics, as supported by prior research [47].

## 4. Conclusions

This research demonstrates the successful development of a novel wearable sweat-sensing platform for real-time monitoring of sodium ions. The microfluidic chip, fabricated using a water-washable resin and 3D printing technology, offers high resolution, rapid fabrication, and excellent hydrophilicity. The We-VoltamoStat potentiostat exhibited high accuracy and precision in measuring electrical currents, with minimal deviations from the reference instrument. The sensor’s current measurements exhibited less than 15% error compared to the Metrohm instrument across the 10 to 200 mM NaCl range. Bland–Altman analysis confirmed a 95% difference interval, while statistical analyses revealed low variation, with a SEM of 1.469 nA, an ICC of 0.998, and a CV of 3.5%. Moreover, the selectivity tests showed minor differences (less than 30 nA) between mixed ions (Na^+^, K^+^, Ca^2+^) and target ions (Na^+^), with a standard deviation below 7. Real-time monitoring during exercise indicated a decrease in sweat sodium levels, with average concentrations ranging from 101 mM to 67 mM over a 50 min exercise session. Future studies could explore the integration of additional sensors to measure other biomarkers in sweat, such as glucose or lactate. Moreover, we also consider integrating flow rate sensing function in the future to simultaneously measure the sweat rate, for example by incorporating the flow rate sensors [48,49]. Additionally, investigating the performance of the proposed device in larger-scale clinical trials could further validate its potential for real-world applications.

## Figures and Tables

**Figure 1 sensors-25-03467-f001:**
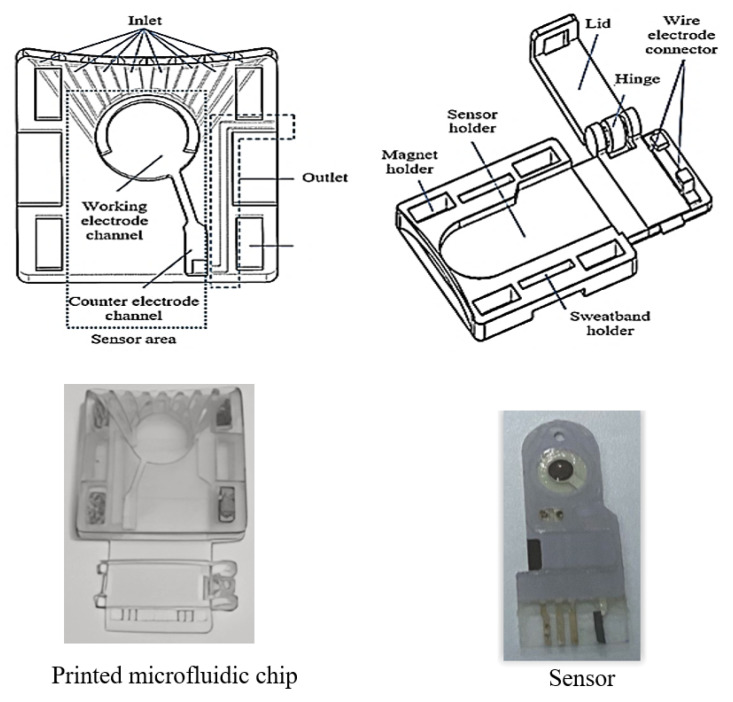
Schematic drawing and 3D print of the microfluidic chip and sodium sensor.

**Figure 2 sensors-25-03467-f002:**
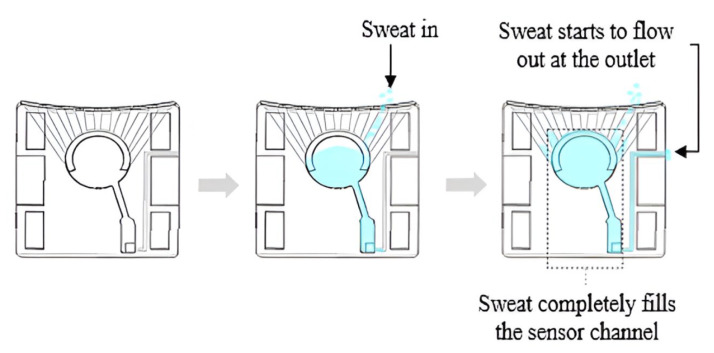
Illustration of sweat flow through the microfluidic chip. The sequential images show sweat entering the sensing channel, filling the sensing area, and flowing out through the outlet, demonstrating the device’s efficiency in fluid transport.

**Figure 3 sensors-25-03467-f003:**
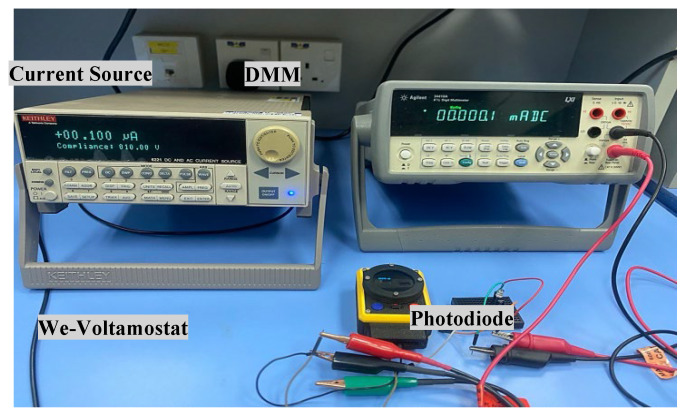
Setup for calibrating the We-VoltamoStat using a Keithley current source. The calibration ensures a reliable measurement by comparing the device’s measurements to a standard instrument across a range of currents.

**Figure 4 sensors-25-03467-f004:**
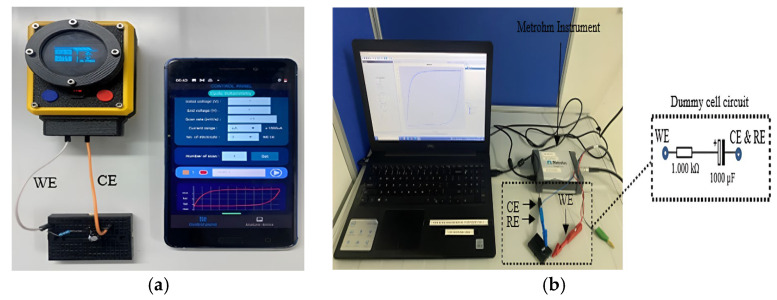
Dummy cell test configurations for validating the We-VoltamoStat. (**a**) Simplified two-electrode setup with the We-VoltamoStat, demonstrating its reliability in measuring electrochemical currents. (**b**) Standard three-electrode setup using the Metrohm instrument.

**Figure 6 sensors-25-03467-f006:**
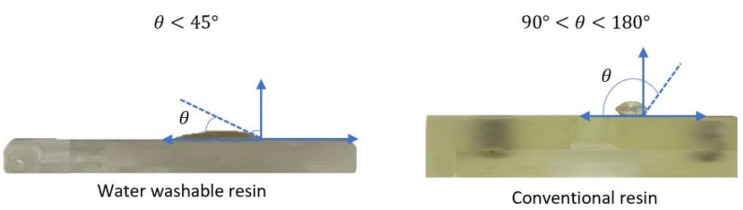
Contact angle measurements of water on microfluidic chip surfaces fabricated with water-washable resin (hydrophilic) and conventional resin (hydrophobic). The results demonstrate the superior hydrophilic properties of the water-washable resin, which enhances sweat flow within the microchannels.

**Figure 7 sensors-25-03467-f007:**
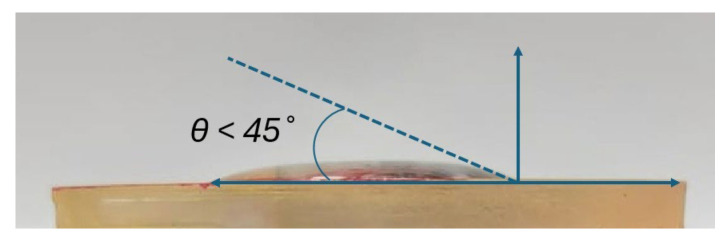
Wettability test of a microfluidic chip after more than two years, showing a contact angle below 45°, indicating that it remains hydrophilic.

**Figure 8 sensors-25-03467-f008:**
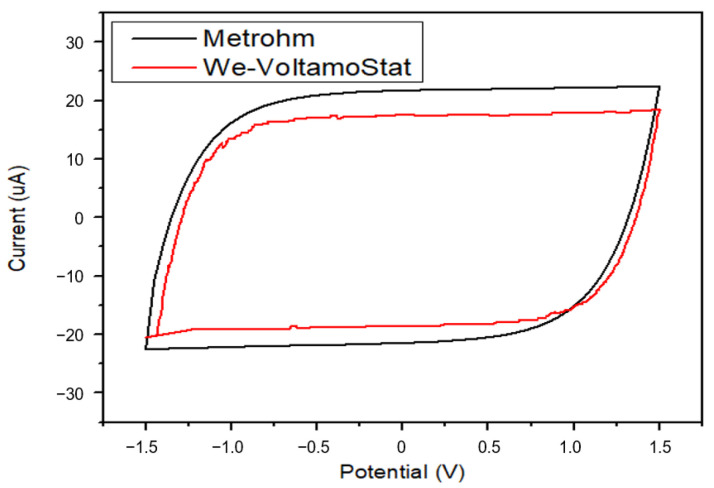
The current-voltage curves generated by the We-VoltamoStat and the Metrohm instrument using a scan rate of 25 mV/s in cyclic voltammetry mode. The nearly identical curves demonstrate the correlation of the We-VoltamoStat with electrochemical measurements.

**Figure 9 sensors-25-03467-f009:**
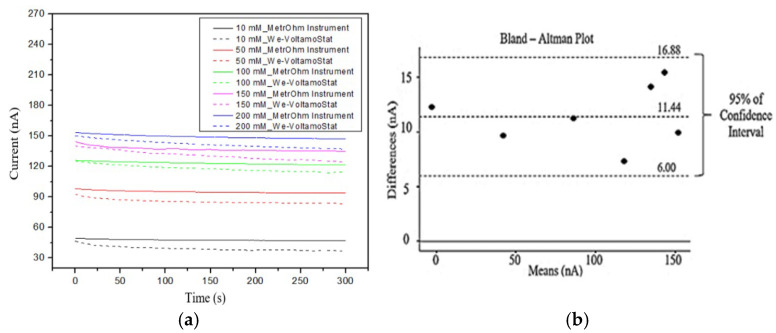
Measurement comparison between the We-VoltamoStat and the Metrohm instruments. (**a**) The graph compares the average currents recorded by the We-VoltamoStat and Metrohm instruments. (**b**) Bland–Altman plots comparing the sodium ion measurements of the across concentrations ranging from 1 mM to 250 mM (each black dot represents an individual sample).

**Figure 10 sensors-25-03467-f010:**
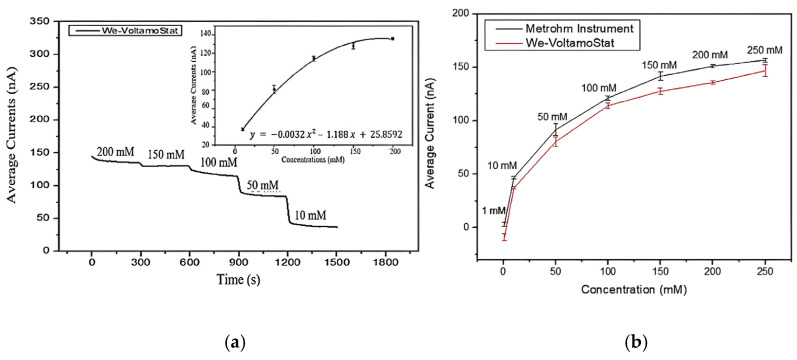
Average current plots for (**a**) sensor response and (**b**) sodium concentration range of 1–250 mM.

**Figure 11 sensors-25-03467-f011:**
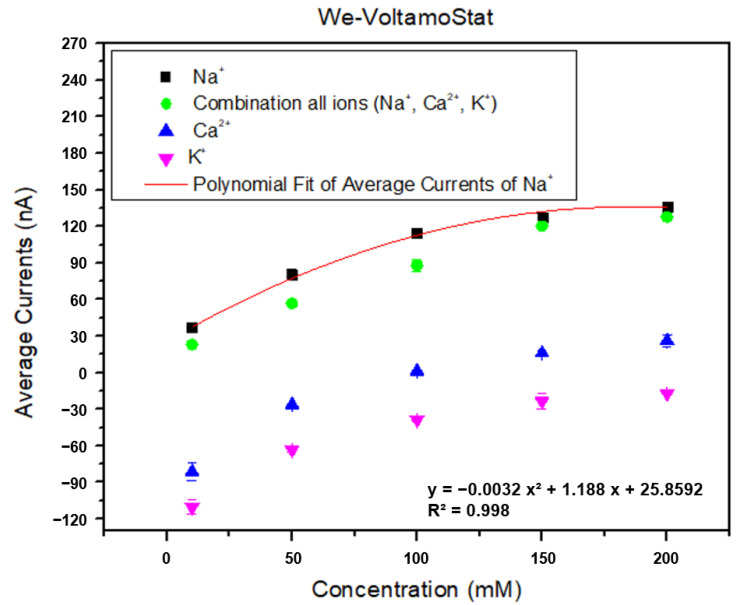
Graphical analysis of selectivity test results for the target ion in the presence of the interference ions.

**Figure 12 sensors-25-03467-f012:**
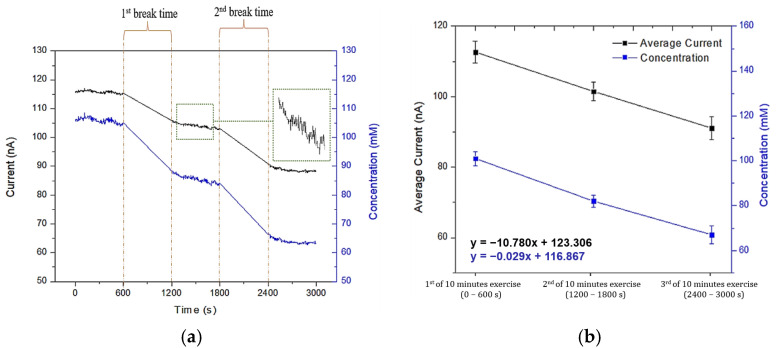
(**a**) Real-time measurement of current (black line) and sodium ion concentration (blue line) during three sessions of 10-min exercise with two rest intervals. (**b**) Averaged values of current (black squares) and sodium concentration (blue circles) during each 10-min exercise session.

**Table 1 sensors-25-03467-t001:** The 3D printer settings used to achieve the chip dimensions.

Setting Parameter	Bottom Layer	Normal Layer	Bottom Retract Speed	Normal Retract Speed
Time Exposure (s)	12	2.5	-	-
Lift Distance (mm)	5	5	-	-
Speed (mm/min)	50	50	150	190

**Table 2 sensors-25-03467-t002:** Sensor output for NaCl concentrations ranging from 1–250 mM.

mM	Current Measurement (nA)		Percentage Error (%)	Mean Difference (nA)
	Metrohm	We-Voltamostat		
	Average			
1	3.252	−9.054	37.8	12.306
10	46.881	37.203	14.6	9.678
50	91.825	80.586	12.2	11.239
100	121.465	114.126	6	7.339
150	141.76	127.624	9.9	14.136
200	151.221	135.78	10.2	15.441
250	156.882	146.95	6.3	9.932

**Table 3 sensors-25-03467-t003:** Validation of the We-VoltamoStat was performed using a paired sample t-test against the Metrohm instrument.

	Metrohm Instrument—We-VoltamoStat
Mean difference (nA)	11.566
SD of differences (nA)	3.285
SEM	1.469
95% CI of difference (nA)	5.128–17.996
ICC	0.998
Sign. (2-tailed)	0.0014
t	7.873
CV (%)	3.518

SD, standard deviation; SEM, standard error of mean; CI, confident interval; ICC intraclass correlation coefficient; t-test statistic; CV, coefficient of variation; sign. (2-tailed); two-tailed probability, *p* < 0.001.

**Table 4 sensors-25-03467-t004:** Mean differences between Na⁺ sensor responses in the presence of interfering ions (Ca²⁺, K⁺, and a combination of ions) at varying concentrations (10–200 mM).

Concentration (mM)	Target Ion (A)	Interference Ions (B)	Mean Differences (A − B)	SEM	SD A	SD B
10 mM	${Na}^{+}$	Combination all ions	14.305	1.7142	1.233	3.428
		${Ca}^{2+}$	118.580	3.5405		4.081
		$K^{+}$	147.548	2.9386		5.877
50 mM	${Na}^{+}$	Combination all ions	23.773	0.4624	4.339	2.813
		${Ca}^{2+}$	106.9358	1.4066		0.925
		$K^{+}$	143.838	2.1695		1.352
100 mM	${Na}^{+}$	Combination all ions	26.442	0.1688	2.613	4.746
		${Ca}^{2+}$	112.956	2.3729		0.338
		$K^{+}$	152.893	1.3065		0.788
150 mM	${Na}^{+}$	Combination all ions	7.294	1.0129	3.182	2.302
		${Ca}^{2+}$	111.592	1.1509		2.026
		$K^{+}$	150.999	3.3026		6.605
200 mM	${Na}^{+}$	Combination all ions	8.222	0.655	1.520	1.937
		${Ca}^{2+}$	109.786	0.968		4.648
		$K^{+}$	152.873	0.760		1.309

## Data Availability

Data are contained within the article.

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
