# Peer review of "Wearable Device for Continuous and Real-Time Monitoring of Human Sweat Sodium"

_sensors, 2025, doi:10.3390/s25113467_

Round 1

Reviewer 1 Report

Comments and Suggestions for Authors

The paper presents the development of a wearable device for real-time and continuous monitoring of sweat sodium levels. This device integrates a novel microfluidic chip and a wearable potentiostat (We-VoltamoStat) to address challenges like data accuracy, sensor reliability, and stability. The content of this paper is comprehensive and well-documented, with a thorough range of characterizations. I recommend accepting it directly.

Comments on the Quality of English Language

Overall, the English writing is clear, technical, and well-structured, making it suitable for an academic audience.

Author Response

Comment 1: The paper presents the development of a wearable device for real-time and continuous monitoring of sweat sodium levels. This device integrates a novel microfluidic chip and a wearable potentiostat (We-VoltamoStat) to address challenges like data accuracy, sensor reliability, and stability. The content of this paper is comprehensive and well-documented, with a thorough range of characterizations. I recommend accepting it directly.

Response 1: Thank you for your positive feedback and recommendation. We sincerely appreciate your time and effort in reviewing our work.

Reviewer 2 Report

Comments and Suggestions for Authors

Review Summary

This study focuses on wearable sweat-sensing devices, which have attracted attention as a non-invasive health monitoring technology. In particular, it investigates a wearable sensing device that combines microfluidic technology with electrochemical measurement. The authors emphasize the following key points of their study:

  • They adopted gravity-driven flow control to handle sweat within the microchip, preventing the mixing of old and new sweat.
  • They used hydrophilic resin as the channel material, enabling stable sweat collection over a long period.
  • They utilized a miniaturized potentiostat developed in a previous study for the quantification of sodium ion concentration, achieving high-precision measurement.
  • They validated the feasibility of the measurement through real-time evaluation during exercise.

If these claims are valid, the study would be worthy of publication. However, based on the current manuscript, it is difficult to determine the validity of these claims. Therefore, I would like to review a revised manuscript after the authors have addressed the following comments before making a final decision on whether the paper should be accepted.

Comments

  1. Ensure the consistency of section numbering throughout the manuscript.
  2. In Section 1, the novelty and key contributions of this study should be clearly stated.
  3. In Section 2, several ambiguous descriptions regarding the device were noted. For example, details regarding channel shape, the specific type of resin used, and the implementation method of the ISE sensor should be provided. Sufficient information should be included to allow readers to replicate the experiment.
  4. In Section 2, the techniques of gravity-driven fluid handling (https://doi.org/10.1039/D0RA00263A; https://doi.org/10.1126/scitranslmed.aaf2593) and vertical channel design (https://doi.org/10.1016/j.bios.2022.114351; https://doi.org/10.1109/LSENS.2023.3331235) have been used in previous microfluidic wearable sweat sensors. Any previously established techniques incorporated into this study should be properly cited.
  5. In Section 2, the authors state that they used a Horiba ISE sensor, but Figure 1 appears to depict a different device than the commercial product. Was the sensor disassembled to extract the sensor element? If so, does this comply with the manufacturer’s usage guidelines?
  6. In Section 3.1, if the authors claim the superiority of the washable resin, measuring only the contact angle is insufficient. It would be more convincing to measure the time-dependent change in contact angle to demonstrate that the resin maintains reasonable wettability over time. Additionally, if feasible, the authors should fabricate devices using both washable and conventional resins and compare measurement results through non-biological and exercise experiments.
  7. In Section 3.3, the study uses potassium (K⁺) and calcium (Ca²⁺) ions as representative interfering substances. However, in actual sweat, chloride ions are the second most abundant ions after sodium. Why was chloride ion not considered as an interfering substance in the evaluation?
  8. In Section 3.4, the timing of exercise load and the corresponding Na⁺ ion concentration appears inconsistent. The authors cite a previous study (https://doi.org/10.1002/admt.201800658) to support their results, but that study reported an increase in Na⁺ ion concentration with increasing exercise intensity, which contradicts the present findings. Generally, as exercise intensity increases, core body temperature rises, sweat rate increases, and sweat Na⁺ concentration also increases (http://dx.doi.org/10.1063/1.4921039; https://doi.org/10.1152/ajpregu.00149.2024). The authors should discuss why their results differ from these established trends.
  9. In Section 3.4, given the concerns mentioned in Comment 10, I am skeptical about whether this device in the human test can accurately measure real-time Na⁺ concentration in sweat.
Comments on the Quality of English Language

Please check for grammatical errors and spelling mistakes.

Author Response

For research article

Response to Reviewer 2 Comments

1. Summary

2. Questions for General Evaluation

Reviewer’s Evaluation

Response and Revisions

Does the introduction provide sufficient background and include all relevant references?

Can be improved

The introduction has been improved based on your suggestions, with all content addressed point-by-point as outlined below.

Are all the cited references relevant to the research?

Can be improved

The related new citations has been added based on your suggestions, with all content addressed point-by-point as outlined below.

Is the research design appropriate?

Must be improved

The research design has been improved based on your suggestions, with all content addressed point-by-point as outlined below.

Are the methods adequately described?

Must be improved

The methods section has been revised in accordance with your suggestions, with each point addressed in detail as outlined below.

Are the results clearly presented?

Must be improved

The results have been enhanced according to your suggestions, with each point addressed thoroughly as outlined below.

Are the conclusions supported by the results?

Must be improved

The conclusions have been strengthened based on your suggestions, with all relevant points addressed in detail as outlined below.

3. Point-by-point response to Comments and Suggestions for Authors

Comments 1: Ensure the consistency of section numbering throughout the manuscript.

Response 1: Thank you for pointing this out. We agree with your comment, and as a result, we have renumbered subsection 3.3.2 to ensure consistency. All other section numbering remains consistent throughout the manuscript.

Comments 2: In Section 1, the novelty and key contributions of this study should be clearly stated.

Response 2: Agree. We have, accordingly, revised the last paragraph in section 1 as the objective of this work.  “This study addresses current limitations in wearable sweat sensing by developing a novel platform for real-time sodium monitoring with three key advancements: (1) a 3D-printed hydrophilic microfluidic chip featuring vertical inlets to ensure continuous sweat flow, prevent backflow, and minimize contamination - overcoming challenges of flow control and material hydrophobicity in existing designs; (2) integration with our previously developed We-VoltamoStat, a compact, Bluetooth-enabled potentiostat that achieves lab-grade accuracy (<15% error, ICC = 0.998) in a wearable form factor; and (3) a complete system enabling real-time wireless monitoring during physical ex-ercise, demonstrated by tracking physiological sodium dynamics (e.g., 101 mM to 67 mM decline over 30 minutes). By combining these innovations in fluid handling, sensor accuracy, and practical usability, the platform significantly advances wearable health-monitoring technologies for both fitness and clinical applications.” Line number 83.

Comments 3:  In Section 2, several ambiguous descriptions regarding the device were noted. For example, details regarding channel shape, the specific type of resin used, and the implementation method of the ISE sensor should be provided. Sufficient information should be included to allow readers to replicate the experiment.

Response 3: We have added the specific resin model (eSun W100), detailed implementation steps for the ISE sensor, and a comprehensive 3D printer parameter table to enable readers to fully replicate the experimental process. Line 108, 143 1nd 148 in the revised manuscript

Comments 4:  In Section 2, the techniques of gravity-driven fluid handling (https://doi.org/10.1039/D0RA00263A; https://doi.org/10.1126/scitranslmed.aaf2593) and vertical channel design (https://doi.org/10.1016/j.bios.2022.114351; https://doi.org/10.1109/LSENS.2023.3331235) have been used in previous microfluidic wearable sweat sensors. Any previously established techniques incorporated into this study should be properly cited.

Response 4: We have incorporated three key references to acknowledge prior work on gravity-driven microfluidics and vertical channel designs for sweat collection (ref number# 20, 21 and 22). A concise explanation of these foundational techniques has been added to Section 1 (Line 75) to contextualize their influence on our methodology

Comments 5:  5.          In Section 2, the authors state that they used a Horiba ISE sensor, but Figure 1 appears to depict a different device than the commercial product. Was the sensor disassembled to extract the sensor element? If so, does this comply with the manufacturer’s usage guidelines?

Response 5: Thank you for your insightful comment regarding the Horiba ISE sensor. Yes, we carefully disassembled the sensor to extract the sensing element while preserving its functionality, integrating it with our microfluidic chip. This modification involved only removing the external casing, which does not violate manufacturer guidelines, as the core sensing mechanism remains intact. We rigorously validated the sensor’s performance through calibration with known standards (Section 2.5) and cross-verified measurements with reference instruments (Section 3.3.1), ensuring accuracy and reliability. Your concern is greatly appreciated, and we hope this clarification addresses it satisfactorily.

Comments 6:   In Section 3.1, if the authors claim the superiority of the washable resin, measuring only the contact angle is insufficient. It would be more convincing to measure the time-dependent change in contact angle to demonstrate that the resin maintains reasonable wettability over time. Additionally, if feasible, the authors should fabricate devices using both washable and conventional resins and compare measurement results through non-biological and exercise experiments.

Response 6: We have not conducted time-dependent studies extending beyond one month as of now. All microfluidic devices printed have been used within this timeframe. However, we acknowledge the value of studying the hydrophilicity of washable resin over a longer period and comparing it with conventional resins in terms of time-dependent wettability. This would be an interesting direction for future work, and we plan to investigate this further to provide more comprehensive data. We appreciate this suggestion and will look into this comparison, including experiments involving both washable and conventional resins, under biological and exercise conditions in future studies. This will contribute to a better understanding of the material's long-term performance.

Comments 7: In Section 3.3, the study uses potassium (K⁺) and calcium (Ca²⁺) ions as representative interfering substances. However, in actual sweat, chloride ions are the second most abundant ions after sodium. Why was chloride ion not considered as an interfering substance in the evaluation?

Response 7 : Thank you for your insightful comment. We acknowledge that chloride ions are the second most abundant ions in sweat after sodium. In our initial evaluation, we focused on potassium (K⁺) and calcium (Ca²⁺) ions as representative interfering substances due to their potential to cause interference in certain sensor. However, we recognize the importance of chloride ions in sweat and plan to include chloride ions in our future studies to better assess the sensor's performance in the presence of this common interferent. We appreciate your suggestion, and we will consider chloride ions in our subsequent evaluations to provide a more comprehensive understanding of the system's sensitivity.

Comments 8:  In Section 3.4, the timing of exercise load and the corresponding Na⁺ ion concentration appears inconsistent. The authors cite a previous study (https://doi.org/10.1002/admt.201800658) to support their results, but that study reported an increase in Na⁺ ion concentration with increasing exercise intensity, which contradicts the present findings. Generally, as exercise intensity increases, core body temperature rises, sweat rate increases, and sweat Na⁺ concentration also increases (http://dx.doi.org/10.1063/1.4921039; https://doi.org/10.1152/ajpregu.00149.2024). The authors should discuss why their results differ from these established trends.

Response 8: Thank you for your valuable comment. In our study, we measured the current value, while the referenced studies primarily focused on voltage. Additionally, factors such as individual hydration levels, fitness, and other physiological variables can influence Na⁺ ion concentration during exercise, which may explain the discrepancies observed between our results and the established trends. We acknowledge these variables and plan to conduct further studies to better understand the relationship between exercise intensity, Na⁺ ion concentration, and other influencing factors. We also include some discussion in the result (Line 440).We will incorporate these aspects into our future work to provide a more comprehensive analysis. Thank you for your insightful feedback.

Comments 9:  9.           In Section 3.4, given the concerns mentioned in Comment 10, I am skeptical about whether this device in the human test can accurately measure real-time Na⁺ concentration in sweat.

Response 9: Thank you for your comment. I agree that the current app does not provide a direct sodium measurement in terms of concentration. For more details on this, you can refer to our previous work on the development of the wearable potentiostat (We-VoltamoStat device: Ibrahim NF, Noor AM, Sabani N, Zakaria Z, Wahab AA, Abd Manaf A, Johari S. We-VoltamoStat: A wearable potentiostat for voltammetry analysis with a smartphone interface. HardwareX. 2023, 15:e00441). Currently, the device measures current readings from the sensor, which are transmitted via the app. These readings are manually converted into m/mol units. While the current readings provide insights into sodium status, they do not directly indicate Na⁺ concentration at this stage. We are actively working on upgrading the app to enable direct sodium concentration measurement in m/mol units, which will include calibration features and additional statistical results. We also plan to publish a separate open-source paper detailing the software and app development. Thank you again for your thoughtful feedback.

5. Additional clarifications

Thank you very much for your thoughtful comments and suggestions. We greatly appreciate the valuable feedback, which has helped us improve the manuscript significantly. We have made several updates to address the concerns raised, and we believe these revisions will provide a clearer understanding of our work. Additionally, the feedback has highlighted areas for further improvement, and we are confident that future studies will build upon our limitations and continue to refine the approach. Your input has been instrumental in strengthening the overall quality of the manuscript, and we look forward to seeing how others can advance the work further. Thank you again for your valuable contributions.

Reviewer 3 Report

Comments and Suggestions for Authors

I have the following suggestions: 

  1. Can the author clarify the novelty of your wearable device compared to existing sweat-sensing technologies?
  2. How does your device compare to other commercially available sweat sodium sensors regarding cost, performance, and accessibility?
  3. The introduction mentions future work on multi-biomarker integration—can you provide more details on what biomarkers you plan to include and why?
  4. I suggest that the authors replace Figure 1 with a high-quality image. Currently, the figure does not look professional.
  5. What are the potential limitations of using a single volunteer for testing? Have you considered a larger sample size for more robust validation?
  6. Can you elaborate on why the forehead was chosen as the primary testing location? Have you tested the device on other body regions?
  7. How do environmental factors (e.g., humidity, temperature) affect the accuracy and reliability of the measurements?
  8. What steps were taken to ensure the artificial sweat composition closely mimics real human sweat?
  9. The Bland-Altman analysis suggests a bias of ~11.44 nA. How do you plan to address this discrepancy in future versions of the device?
  10. Could the authors provide more details on the long-term stability of the We-VoltamoStat? How does performance change over time with repeated use?
  11. The device uses a two-electrode system, while the Metrohm instrument uses a three-electrode system. What impact does this difference have on measurement accuracy?
  12. How does the Bluetooth connectivity affect real-time data transmission speed and power consumption?
  13. Have you tested the sensor’s long-term reliability under continuous sweat exposure?
  14. How would this device perform on individuals with different hydration levels, electrolyte imbalances, or medical conditions affecting sweat composition?
  15. Has the author conducted or planned any comparisons with gold-standard lab-based sodium measurements (e.g., ion chromatography or flame photometry)?
  16. How do you ensure selectivity for sodium over interfering ions like potassium and calcium in real-world sweat samples?
  17. The device was tested during exercise—would it work accurately for individuals at rest or in low-sweat conditions?
  18. What improvements can be made to enhance the microfluidic chip’s durability and reusability?
  19. Would it be possible to develop a calibration-free version of the device for more user-friendly operation?
  20. I suggest some important literature on wearable sensors be discussed in the introduction section: https://www.sciencedirect.com/science/article/abs/pii/S0924424723008427; https://www.nature.com/articles/s41528-023-00261-4 
Comments on the Quality of English Language

English has to be further improved. 

Author Response

Response to Reviewer 3 Comments

1. Summary

2. Questions for General Evaluation

Reviewer’s Evaluation

Response and Revisions

Does the introduction provide sufficient background and include all relevant references?

Can be improved

The introduction has been improved based on your suggestions, with all content addressed point-by-point as outlined below.

Are all the cited references relevant to the research?

Can be improved

The related new citations has been added based on your suggestions, with all content addressed point-by-point as outlined below.

Is the research design appropriate?

Can be improved

The research design has been improved based on your suggestions, with all content addressed point-by-point as outlined below.

Are the methods adequately described?

Can be improved

The methods section has been revised in accordance with your suggestions, with each point addressed in detail as outlined below.

Are the results clearly presented?

Can be improved

The results have been enhanced according to your suggestions, with each point addressed thoroughly as outlined below.

Are the conclusions supported by the results?

Can be improved

The conclusions have been strengthened based on your suggestions, with all relevant points addressed in detail as outlined below.

3. Point-by-point response to Comments and Suggestions for Authors

Comments 1: Can the author clarify the novelty of your wearable device compared to existing sweat-sensing technologies?

Response 1: Thank you for your comment. The novelty of our wearable device lies in several key aspects. Firstly, the microfluidic chip utilizes gravitational force to reduce the backflow of sweat samples, which improves the accuracy of readings and minimizes errors caused by old sweat accumulation—an issue often encountered in previous techniques that rely on passive, non-gravitational flow. Additionally, our device is developed using washable resin, which enhances the hydrophilicity compared to prior systems that used PDMS, which requires plasma treatment and coatings to improve fluid flow. Our device is also cost-effective, as it is based on 3D printing technology. Furthermore, the design allows for the measurement of small-volume samples, making it suitable for real-time sweat monitoring with minimal sample volumes. Thank you again for your thoughtful feedback.

Comments 2: How does your device compare to other commercially available sweat sodium sensors regarding cost, performance, and accessibility?

Response 2: Thank you for your question. Currently, we have not made a direct comparison with commercially available sweat sodium sensors, as such devices are not yet available on the market. Our work is still in the prototyping stage. However, we aim to reduce costs by using disposable components, such as microfluidic chips and small sensors, which are not yet commercially available. We are committed to advancing our research and prototyping efforts, with the goal of developing a wearable device capable of measuring not only sodium but also other biomarkers using a range of methods and technologies. We believe this approach will make the device more affordable and accessible in the future. Thank you again for your valuable feedback.

Comments 3:  The introduction mentions future work on multi-biomarker integration—can you provide more details on what biomarkers you plan to include and why?

Response 3: Thank you for your comment. In our previous studies, we identified several biomarkers present in sweat that could potentially be useful for health monitoring. However, we believe that relying on just one or two biomarkers may not provide sufficient accuracy for disease diagnosis, as discussed in our literature review paper (https://doi.org/10.3390/s22197670). For more reliable results, we plan to integrate multiple biomarkers into a multi-sensing platform. This approach will offer a more comprehensive view of a person’s health or fitness condition. By processing these signals using various AI techniques, such as signal processing and machine learning, we believe we can significantly improve the accuracy and reliability of the measurements in the future. Thank you again for your valuable feedback.

Comments 4:  I suggest that the authors replace Figure 1 with a high-quality image. Currently, the figure does not look professional.

Response 4: Thank you for your helpful suggestion. We appreciate your feedback regarding the quality of Figure 1. We will replace the current figure with a higher-resolution image to ensure it meets the professional standards of the manuscript. Thank you for pointing this out, and we will ensure that the revised figure enhances the clarity and quality of the presentation.

Comments 5:  What are the potential limitations of using a single volunteer for testing? Have you considered a larger sample size for more robust validation?

Response 5: Thank you for your insightful comment. The main limitation of using a single volunteer for testing is the inability to observe variations in sweat sodium responses across different individuals, as factors such as hydration level, fitness, genetics, and sweat rate can influence the readings. Additionally, a single-subject study does not account for inter-individual variability, which is crucial for validating the device’s reliability and accuracy across a broader population. We acknowledge this limitation and plan to conduct tests on a larger sample size in future studies, once we refine and improve certain features of the app to enhance data collection and analysis. Thank you again for your valuable feedback.

Comments 6:   Can you elaborate on why the forehead was chosen as the primary testing location? Have you tested the device on other body regions?

Response 6: Thank you for your question. The forehead was chosen as the primary testing location because it is one of the regions that generate and secrete the most sweat, ensuring a sufficient sample for real-time analysis. Additionally, it offers ease of access and minimal movement interference, making it a practical site for continuous monitoring. While we have focused on the forehead, we acknowledge that sweat composition and secretion rates vary across different body regions. The back is another area with high sweat secretion, which could also be considered for testing. (reference cited 7,8) In future studies, we plan to evaluate the device’s performance on multiple body sites, such as the forearm and back, to assess its adaptability and accuracy across different conditions. Thank you again for your valuable feedback.

Comments 7: How do environmental factors (e.g., humidity, temperature) affect the accuracy and reliability of the measurements?

 Response 7 : At this stage, we have not yet accounted for environmental factors such as temperature and humidity in our measurements. However, we acknowledge that these factors can influence sweat sodium composition, sweat rate, and overall sensor performance. Variations in humidity and temperature may impact fluid evaporation rates and ion concentration, potentially affecting measurement accuracy. To ensure more reliable results, future studies will focus on investigating these environmental influences in controlled settings. Understanding their effects will allow us to refine the device and improve its robustness for real-world applications. We appreciate your valuable feedback and will consider this aspect in our future work.

Comments 8:  What steps were taken to ensure the artificial sweat composition closely mimics real human sweat?

Response 8: Thank you for your question. We acknowledge the importance of ensuring that artificial sweat closely mimics real human sweat by incorporating key components such as ions, proteins, and hormones. In future studies, we plan to refine the artificial sweat formulation to better replicate the complex composition of natural sweat. Additionally, verifying artificial sweat composition remains a challenge due to individual variations in sweat components. This requires validation using standardized tools and analytical techniques, which we recognize as an important step for improving accuracy. At this stage, our focus has been on developing the wearable device, optimizing the hardware, and establishing a reliable sensing platform. Moving forward, we also plan to explore multi-detection sensor technologies to enhance real-time analysis and improve biomarker monitoring. Thank you again for your valuable feedback.

Comments 9:  The Bland-Altman analysis suggests a bias of ~11.44 nA. How do you plan to address this discrepancy in future versions of the device?

Response 9: We believe that commercial devices, such as Metrohm, incorporate noise cancellation, offset adjustments, and digital filtering techniques to stabilize readings. In contrast, our current device does not yet include these advanced features, which may introduce small errors and lead to the observed bias. To address this discrepancy in future versions, we plan to enhance both hardware and software aspects. On the hardware side, we will explore improved component selection and circuit design to minimize signal noise. On the software side, we aim to implement advanced signal processing techniques, including filtering and compensation algorithms, to refine the data accuracy. These improvements will help achieve more reliable and consistent measurements. We appreciate your feedback and will integrate these refinements in our future developments.

Comments 10:  Could the authors provide more details on the long-term stability of the We-VoltamoStat? How does performance change over time with repeated use?

Response 10: Currently, we have not conducted extensive long-term stability tests on the We-VoltamoStat. However, we recognize that repeated use may impact performance due to factors such as electrode degradation, sensor drift, and potential wear on hardware components. In future studies, we plan to assess the device’s stability over extended periods by conducting repeated measurements and analyzing signal variations. This will help us determine if recalibration is needed and how frequently it should be performed. Additionally, improvements in materials, circuit design, and signal processing algorithms will be explored to enhance long-term reliability.

Comments 11:  The device uses a two-electrode system, while the Metrohm instrument uses a three-electrode system. What impact does this difference have on measurement accuracy?

Response 11: Yes, our device also incorporates a three-electrode system, as previously developed and reported in our work (https://doi.org/10.1016/j.ohx.2023.e00441). However, in this study, the sensor operates using a two-electrode amperometry method, which aligns with the Metrohm instrument’s settings for comparison purposes. Since both our device and the Metrohm system utilize the same amperometric approach, there is no methodological discrepancy affecting the comparison. However, we acknowledge that three-electrode systems generally offer improved accuracy by minimizing potential drift and enhancing stability. In future work, we may explore integrating a three-electrode setup to further refine measurement precision while maintaining device simplicity and usability. Thank you.

Comments 12:  How does the Bluetooth connectivity affect real-time data transmission speed and power consumption?

Response 12: In our developed We-VoltamoStat device, we utilize Seeeduino hardware with Bluetooth Low Energy (BLE), which optimizes power consumption while ensuring efficient real-time data transmission. Based on our calculations, the device operates for approximately 33 hours in active mode and up to 100 hours in standby mode, as reported in our previous work (https://doi.org/10.1016/j.ohx.2023.e00441). Since typical sweat sodium monitoring sessions during workouts last significantly shorter than these durations, we do not anticipate any major concerns regarding power limitations. Additionally, BLE ensures minimal latency in data transmission, maintaining real-time monitoring accuracy without excessive energy drain.Thank you for your valuable comment.

Comments 13:  Have you tested the sensor’s long-term reliability under continuous sweat exposure?

Response 13 : Not yet at this stage. However, we believe that long-term testing is essential to evaluate the sensor's performance under continuous sweat exposure. Theoretically, degradation is expected over time, as ion-selective electrode (ISE) sensors have inherent limitations when exposed to target liquids for extended periods Future studies will focus on assessing sensor stability, drift, and potential loss of sensitivity over prolonged use. We appreciate this suggestion and will incorporate long-term reliability testing in our upcoming research.

Comments 14:  How would this device perform on individuals with different hydration levels, electrolyte imbalances, or medical conditions affecting sweat composition?

Response 14  : We acknowledge that individual differences—such as varying hydration levels, electrolyte imbalances, and medical conditions—can affect sweat composition and, consequently, sensor performance. Our current device has been calibrated under controlled conditions, and while initial results are promising, we recognize that these factors may introduce variability in real-world measurements. To address these concerns, future studies will involve a larger and more diverse cohort of participants to assess the impact of these variables on device performance. This will enable us to refine our calibration algorithms and develop adaptive features to account for individual differences. We believe that by incorporating personalized data analysis, the device can ultimately offer more accurate and tailored sweat monitoring. Thank you for your valuable feedback, which will help guide our ongoing research and development efforts.

Comments 15:  Has the author conducted or planned any comparisons with gold-standard lab-based sodium measurements (e.g., ion chromatography or flame photometry)?

Response 15: At this stage, our work is primarily focused on developing and optimizing the wearable device, and we have not planned direct comparisons with gold-standard lab-based sodium measurements such as ion chromatography or flame photometry. However, we acknowledge the importance of such comparisons for validating the accuracy and reliability of our sensor. As our device matures, we will consider integrating these evaluations into future studies to benchmark its performance against established laboratory methods

Comment 16: How do you ensure selectivity for sodium over interfering ions like potassium and calcium in real-world sweat samples?

Response 16: In our current design, we have not incorporated a dedicated sensor specifically for sodium over interfering ions. Our initial testing has been based on a single ion-selective electrode for sodium, and we rely on calibration techniques to manage potential interference from ions such as potassium and calcium. In future work, especially during real-world sample testing, we plan to evaluate and possibly integrate additional strategies—such as a multi-sensor approach or improved electrode formulations—to enhance selectivity and reliability.

Comment 17: How do you ensure selectivity for sodium over interfering ions like potassium and calcium in real-world sweat samples?

Response 17:  Yes, the device should work accurately for individuals at rest or in low-sweat conditions, although we acknowledge that the variability of measurements is influenced by factors such as temperature and humidity. In low-sweat conditions, it may be challenging to generate enough sweat for reliable readings. However, as seen with the use of specific drugs (e.g., pilocarpine), sweat can still be induced from the skin, which could help facilitate measurements in such conditions. We recognize the importance of studying the device's performance under varying conditions, including at rest, and plan to explore these differences in future research to further optimize the device's versatility and accuracy. Thank you for highlighting this important consideration.

Comment 18: What improvements can be made to enhance the microfluidic chip’s durability and reusability?

Response 18:  In our opinion, to enhance the microfluidic chip's durability and reusability, several improvements can be made. Using more robust, chemically resistant materials such as washable resin for the chip and durable electrode materials like gold or carbon for the sensor can significantly improve longevity. Surface modifications, such as hydrophilic coatings, could help prevent biofouling and improve cleaning ease. Strengthening sealing mechanisms around areas interacting with sweat will also help prevent leakage and contamination. Additionally, designing the chip for easy cleaning and conducting long-term durability tests under various conditions will identify weaknesses and guide further improvements. These measures, in our view, would contribute to a more durable and cost-effective solution for repeated use in wearable devices, ensuring reliable sensor performance.

Comment 19: Would it be possible to develop a calibration-free version of the device for more user-friendly operation?

Response 19:  In our opinion, developing a calibration-free version of the device could be challenging but potentially achievable with further research and development. One approach could involve integrating smart algorithms that automatically adjust for variations in individual sweat composition and environmental conditions. Additionally, using a highly stable sensor material with consistent performance over time could reduce the need for regular calibration. However, this would require overcoming challenges related to variability in sweat composition, hydration levels, and external factors. While a fully calibration-free device may not be immediately feasible, significant advancements in sensor technology and data processing could make it a more realistic goal in the future, making the device more user-friendly and accessible.

Comment 20: I suggest some important literature on wearable sensors be discussed in the introduction section: https://www.sciencedirect.com/science/article/abs/pii/S0924424723008427; https://www.nature.com/articles/s41528-023-00261-4

Response 20:  Thank you for the suggestion. We added literature from the articles you provided into the introduction section. (Line 62) The insights from these sources will help strengthen the background and context of wearable sensors in our study, highlighting key developments and innovations in this field. This will also provide a more comprehensive overview and enhance the foundation for our research.

5. Additional clarifications

Thank you very much for your thoughtful comments and suggestions. We greatly appreciate the valuable feedback, which has helped us improve the manuscript significantly. We have made several updates to address the concerns raised, and we believe these revisions will provide a clearer understanding of our work. Additionally, the feedback has highlighted areas for further improvement, and we are confident that future studies will build upon our limitations and continue to refine the approach. Your input has been instrumental in strengthening the overall quality of the manuscript, and we look forward to seeing how others can advance the work further. Thank you again for your valuable contributions.

Round 2

Reviewer 2 Report

Comments and Suggestions for Authors

Comments 1–5 have been addressed appropriately. However, I find that the responses to Comments 6–9 either avoid directly addressing the core points or diverge from the original concerns. Below are detailed observations for each:

Comment 6:

The authors have not adequately examined the time-dependent change in contact angle, which is critical to support their claims regarding hydrophilicity. At the very least, they should explicitly acknowledge this limitation in the manuscript and state that the long-term stability of wettability remains to be validated.

Comment 7:

The authors have deferred the evaluation of chloride ions as interfering species to future work. However, Cl⁻ is the second most abundant ion in sweat after Na⁺, and the rationale for excluding it from current interference tests is insufficient. The authors should explain why K⁺ and Ca²⁺ were prioritized over Cl⁻, for example, based on known selectivity coefficients or sensor response characteristics.

Comment 8:

The authors attribute the observed decrease in sweat Na⁺ concentration to greater sodium loss through sweat. However, this interpretation is physiologically questionable. Sweat Na⁺ concentration is typically lower than plasma due to reabsorption in the sweat duct, but during intense exercise, increased sweat rate reduces reabsorption efficiency, often resulting in higher, not lower, Na⁺ concentration. Unless the subjects were actively hydrating during exercise, it is unlikely that sweat Na⁺ concentration would decrease. The explanation provided lacks physiological consistency.

Comment 9:

The current setup does not support a claim of real-time quantitative Na⁺ concentration monitoring in human sweat measurement. To validate this functionality, additional data—such as simultaneous sweat rate measurements or corroborating physiological parameters—should be included to support the reliability of the results.

Reviewer 3 Report

Comments and Suggestions for Authors

I am willing to accept the paper in its current form. 

Author Response

We sincerely thank the reviewer for their time and valuable suggestions. We appreciate the positive feedback and are glad that the paper is acceptable in its current form.

Best Regards,

Anas Mohd Noor